# Investigating the Factors Influencing the Adoption of Blockchain Technology across Different Countries and Industries: A Systematic Literature Review

**Agostino Marengo** [1],[*] and **Alessandro Pagano** [2]

1   Department of Human Science, University of Foggia, 71122 Foggia, Italy
2   Department of Economics, University of Bari, 70121 Bari, Italy; alessandro.pagano@uniba.it
*   Correspondence: agostino.marengo@unifg.it

**Abstract:** Despite the reported disruptive nature of blockchain technology in the extant literature, its adoption is slower than its potential. This difference between the technology's promises and its current adoption has sparked interest in understanding the factors impeding widespread adoption. This systematic literature review (SLR), drawn from 1786 studies published between 2008 and May 2023, seeks to address this gap. Specifically, our research explores the influence of factors and their differences and commonalities on blockchain adoption. The SLR, examining individual and organisational perspectives, identifies 152 unique factors influencing 25 industries across 21 countries. This review also highlights distinct commonalities and variations in these factors across industries and countries. For instance, while regulatory issues and costs were universal concerns, the importance of technical understanding diverged between industries. Furthermore, country-specific factors, including local regulations and cultural aspects, emerged as significantly influenced insights that provide a comprehensive perspective on the dynamics of blockchain adoption, offering valuable guidance to industry practitioners and researchers striving to navigate the complexities of blockchain integration.

**Keywords:** blockchain; adoption; factors; countries; literature; industries





## 1. Introduction

The Industrial Revolution brought about several emerging technologies, leading to significant changes across many industries. Among these technologies, blockchain technology (BCT) has captured the attention of many, becoming one of the most unique, disruptive technologies of the 21st century. Introduced initially in the form of Bitcoin by Satoshi Nakamoto [1], blockchain has shown to be a vast and multifaceted technology, with Bitcoin representing just one of many possible applications [2].

Blockchain is a distributed and decentralised technology that uses cryptographic measures to securely store data in interconnected blocks, forming a transparent, immutable, and decentralised network [3]. The advent of this technology has presented the world with new methods of transaction and data management that are expected to revolutionise conventional processes.

The inherent characteristics of blockchain, such as its decentralised nature, fewer intermediaries, proof of work (POW), proof of stake (POS), cryptographic security, audibility, and near real-time update capabilities, present a significant shift from traditional centralised systems [4,5]. Furthermore, the types of blockchains, like permissioned, permissionless, centralised, decentralised, and hybrid, propose profound implications on how trust, accountability, and efficiency are established in systems involving peer-to-peer transactions [6]. They are also why blockchain technology holds such tremendous potential for transforming operations across various industries, from finance to healthcare, supply chain management, and beyond [7–9].

Despite this broad applicability and potential, blockchain technology's actual adoption and application exhibit significant variation across industries and countries [10].

Previous research into this field has primarily been industry specific or country specific, limiting our understanding of blockchain adoption from a global and cross-sectoral perspective [11,12]. This fragmented knowledge, combined with the nascent state of technology, has created a research gap, emphasising the need for a more comprehensive, cross-sectoral, and global study.

This knowledge gap is significant given the scale of investments flowing into blockchain technologies. Between 2009 and 2018, over USD 13.1bn was invested in blockchain startups, with the USA, China, and the European Union being the most prominent contributors [13–15]. While these figures speak to the increasing recognition of the technology's potential, they also underscore the critical need for an improved understanding of the factors influencing its adoption.

Past studies have identified key factors influencing blockchain adoption, such as perceived trust, perceived usefulness, and organisational readiness [16,17]. These findings have significantly contributed to understanding the factors driving blockchain adoption decisions. Moreover, many studies have employed robust research methodologies, including surveys, case studies, and experiments, to provide empirical evidence supporting their findings [18–20]. This empirical basis enhances the validity and reliability of the research outcomes. However, the existing body of the literature has certain limitations and weaknesses. One notable limitation is the lack of generalizability of findings due to the narrow focus of many studies on specific industries or countries. While these studies offer valuable insights within their respective contexts, it is imperative to consider a broader range of industries and countries to ensure a comprehensive understanding of blockchain adoption across diverse settings [20,21].

Additionally, some studies have heavily relied on self-reported data, which may introduce response biases or subjective interpretations of the factors influencing blockchain adoption. To minimise the biases, explicit and systematic methods can be used by reviewing articles and all available evidence. This leads to reliable findings from which conclusions can be drawn and decisions made [22]. Therefore, this paper offers a systematic review of the literature exploring the factors influencing blockchain adoption across various industries and countries. Unlike past studies that merely identify the factors, this study also explains how the factors influence blockchain adoption. Specifically, the purpose of this study is to find answers to the following research questions:

**RQ 1:** *How do the factors influence blockchain adoption across industries and countries?*

**RQ 2:** *How do the commonalities and differences influence blockchain adoption across different industries and countries?*

The rest of this paper proceeds as follows: Section 2 outlines the materials and methods, Section 3 presents the results, Section 4 describes the discussion, and Section 5 presents this study's conclusions, limitations, and future directions.

## 2. Materials and Methods

To answer the research questions, this present study conducted a systematic literature review to identify the factors influencing the adoption of blockchain technology across different countries and industries. This study followed the guidelines outlined by Okoli [23] and Kitchenham and Brereton [24]. The review process was conducted in stages: search strategy, inclusion and exclusion criteria, screening and selection, and data extraction and analysis. Here, we provide further details for each stage.

### 2.1. Search Strategy

A comprehensive search of electronic databases was conducted for peer-reviewed articles published between 2009 (the year of the introduction of blockchain) and May 2023. This study chose widely used databases in information systems research, which include, Table 1,

Google Scholar, ScienceDirect, Web of Science, Scopus, IEEE Xplore, Springer, Emerald, and the ACM Digital Library. This study used multiple databases because searching multiple databases can maximise available data and consider all relevant literature. According to Ewald and Klerings [25], searching two or more databases decreased the risk of missing relevant studies.

**Table 1.** Number of articles retrieved from databases.

| Database | Number of Articles |
| --- | --- |
| Scopus | 51 |
| IEEE Xplore | 7 |
| Springer | 656 |
| Web of Science | 35 |
| Google Scholar | 1020 |
| Emerald | 5 |
| ACM | 2 |
| Science Direct | 7 |
| Total | 1783 |

The search terms combined relevant keywords about blockchain technology, e.g., "blockchain", "distributed ledger technology", "adoption", "diffusion", acceptance", "factor", "determinants", and "elements", using Boolean operators like AND, OR. The search was performed in the articles' titles, abstracts, and keywords. The following table shows the number of papers retrieved from each database.

We used advanced research; Springer and Google Scholar returned the most irrelevant papers.

### 2.2. Inclusion and Exclusion Criteria

Specific inclusion and exclusion criteria were applied to ensure the selected articles' quality and relevance. Only articles that met the following criteria were included in this review:

*Written in English:* Since English is the primary language of scientific communication, articles written in English were considered for comprehensively analysed text articles and were preferred to ensure a comprehensive analysis of the factors influencing blockchain adoption. We refer to full-text articles that have complete content and open-access articles. This includes open-access articles and articles that may require a subscription or access to academic libraries.

*Empirical investigation:* Only articles that empirically investigated the factors influencing the adoption of blockchain technology were included. This criterion aimed to focus on studies that provided improvidence and insights.

*Published in journal and conference papers:* Articles published in reputable journals and conference proceedings are included to capture diverse research outputs.

Articles were excluded if they met any of the following criteria:

*Non-English language:* Articles written in languages other than English were excluded due to language limitations.

*Focus on cryptocurrency without a clear link to blockchain technology:* Articles solely focused on cryptocurrency without a clear connection to blockchain technology were excluded, as the objective was to focus on factors specific to blockchain adoption.

*Lack of specific address to adoption factors:* Articles that did not specifically address the factors influencing blockchain adoption were excluded to ensure the relevance and focus of this review.

### 2.3. Screening and Selection

We employed a systematic process for identifying and removing duplicates. Initially, we used EndNote 20 software to remove exact duplicates based on title and author information. Subsequently, we manually reviewed the remaining articles to ensure that no

duplicates were overlooked. The remaining articles were then selected for a full-text review based on the inclusion and exclusion criteria. Any disagreements about inclusion were resolved via discussion until consensus among authors was reached.

*2.4. Data Extraction and Analysis*

Data extraction was conducted from the included articles to capture key information, including the country, industry, and key findings related to the adoption of blockchain technology. The factors influencing blockchain adoption were identified and categorised into main groups based on recurring themes and patterns, as discussed in Section 3.2.

To ensure the rigour of our systematic review, we followed the PRISMA (Preferred Reporting Items for Systematic Reviews and Meta-Analyses) guidelines as incorporated in the review on blockchain technology by Sahoo, Kumar [26]. This measure enhanced the reliability and validity of the findings [27]. In light of PRISMA guidelines, Figure 1 demonstrates the number of articles included in this study.

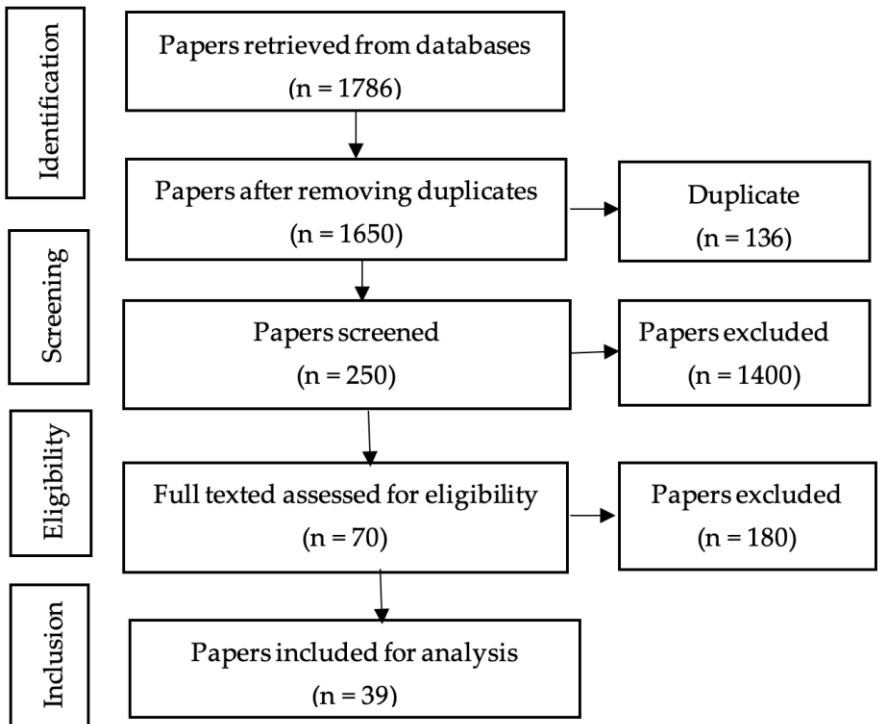

**Figure 1.** PRISMA flow chart for this study.

As shown in Figure 1, 39 articles from 1786 are included in this study. The reduction occurred due to a rigorous screening and selection process based on specific inclusion and exclusion criteria. Initially, 136 duplicate papers were removed, resulting in 1650 unique papers. These papers were then subjected to a thorough review, excluding 1400 papers that were deemed irrelevant or did not meet the criteria. The remaining 250 papers were further screened, excluding 180 papers that lacked methodological rigour or relevance to the research objectives. Finally, the remaining 70 papers underwent a comprehensive analysis, including 39 papers that provided substantial insights into the factors influencing blockchain adoption.

## 3. Results

This section presents detailed results of our analysis of the factors influencing blockchain adoption across different industries and countries. We organise the results into different tables based on the research questions.

**RQ 1:** *What factors influence blockchain adoption across different industries and countries?*

### 3.1. Factors for Industries and Countries

Drawing upon 39 diverse studies, the analysis provides a comprehensive understanding of factors influencing the adoption of blockchain technology. These studies cover a wide range of 25 distinct industries and 21 countries and report 152 unique factors. The most frequently studied industry appears to be the "supply chain", which underscores the significance of blockchain in enhancing transparency and efficiency in this industry. "India" surfaces as the country with the most studies, reflecting the growing interest and application of blockchain technology in this country. Regarding influencing factors, common themes across the studies include "perceived ease of use", "perceived usefulness", and "top management support", all of which indicate the crucial role of user perception and organisational backing in facilitating blockchain adoption. Table 2 summarises the factors identified in this SLR for different industries and countries.

**Table 2.** Factors affecting the adoption of blockchain technology.

| Source | Factors | Industry | Country |
|---|---|---|---|
| [28] | Perceived trust, perceived ease of use, autonomous motivation, perceived usefulness | Taxing System | Bangladesh |
| [29] | Perceived ease of use, government support, vendor support, adoption intention, perceived usefulness, security concerns, top management support, technology readiness, the complexity of technology, technology compatibility, relative advantage, cost concerns | Supply chain | India |
| [30] | Optimism, innovativeness, discomfort, insecurity, perceived ease of use, perceived usefulness | Intelligence communities | Malaysia |
| [31] | Perceived ease of use, output quality, trust, perceived usefulness, information quality | crowdsourcing platform | China |
| [32] | Disintermediation, traceability, trust, coordination/control, compliance, price of technology products/services | Agriculture food supply chain | India |
| [33] | Perceived usefulness, trialability, relative advantage, compatibility, perceived ease of use | Education | Malaysia |
| [34] | Technology characteristics, task characteristics, inter-organisational trust, technology trust, user satisfaction, service quality, information quality, system quality, intention to adopt blockchain, blockchain efficiency, social influence, facilitating conditions, efforts expectancy, performance | Supply chain | Australia |
| [35] | Customer satisfaction, cost saving, favourable economy, increased use of technological devices, government support | Insurance | Malta |

**Table 2.** *Cont.*

| Source | Factors | Industry | Country |
|---|---|---|---|
| [36] | Transparency, smart contracts, shared database, secured database, reduced settlement lead times, reduced transaction cost, improved risk management, decentralised database, auditability, privacy, anonymity, immutability, provenance, traceability | Supply chain | India |
| [37] | Perceived usefulness, individual technology fit, task technology fit, perceived safety, network externality, perceived ease of use | Logistics | Taiwan |
| [38] | Strategic orientation, social influence, innovativeness, perceived ease of use, perceived usefulness, self-efficacy, complexity, security | Tourism | Taiwan |
| [39] | Facilitating conditions, performance expectancy, social influence, effort expectancy, trust | Supply Chain | Brazil |
| [40] | Regulatory support, competitive pressure, market dynamics, cost, upper management support, complexity, relative advantage | Supply chain | Malaysia |
| [41] | disintermediation, relative advantage, maturity, smart contact coding, data security, compatibility, complexity, perceived benefits, blockchain knowledge, participation incentives, innovativeness, technology readiness, business model readiness, organisational size, top management support, organisational readiness, critical user mass, trading partner support, business use cases, government support, industry pressure, market dynamics, regulatory environment | SMEs | Ireland |
| [42] | Increase in data availability, reduction in information, asymmetry, easy verification of transactions, comprehensibility of the transaction, data accuracy and reliability, data inalterability, exclusion of false information from contractual information, hacking attempts system denials, high-security encryption, cost reduction via the exclusion of intermediaries, contract conclusion with a reasonable fee, cost reduction due to process efficiency | Real Estate | Kosovo |
| [43] | Sufficient capital, staff training, support from the senior management, ease of local legislation, support from the shipping community, professional consultation, and assistance | Maritime | Singapore |

**Table 2.** *Cont.*

| Source | Factors | Industry | Country |
|---|---|---|---|
| [44] | Complexity, ease of use, lack of interoperability and standardisation, lack of scalability and system speed, huge resource (energy, infrastructure), initial capital requirement, lack of government regulation, lack of trust among agro-stakeholders | Agriculture | India |
| [21] | Cost, governance, perceived compatibility, perceived ease of use, perceived usefulness, privacy, observability, security, trialability, people's readiness, process readiness, technology readiness, top management enthusiasm, top management expertise, top management support, competitive pressure, customer's influence, connection with ICT providers | Supply chain | Europe |
| [45] | Budget availability, financial risk and uncertainty, cost saving, talent and knowledge acquisition, stakeholder's awareness and acceptance, blockchain ecosystem, disintermediation and business process, infrastructure and platform integration, standardisation, security and privacy, blockchain maturity and use case, management support, training and skills, HIT strategy, regulation compliance, regulatory uncertainty and governance, incentives availability | Health | USA |
| [46] | Regulatory governance and industry standards, technological improvements, and optimisation on efficiency, tracking and tracing, digitalised management, air traffic management | Aviation | Republic of Korea |
| [47] | Relative advantage, upper management support, human resources, compatibility, cost, complexity, technological infrastructure, and architecture. | Supply Chain | Sri Lanka |
| [48] | Perceived benefits, complexity, compatibility; organisational readiness, top management support, organisational size, regulatory environment, market dynamics, transparency, integrity of data, immutability | Government organisations | Malaysia |
| [49] | Organisational readiness, trading partner pressure, perceived benefits, complexity, top management support, compatibility | SMEs | South Africa |

**Table 2.** *Cont.*

| Source | Factors | Industry | Country |
|---|---|---|---|
| [50] | trialability, relative advantage, competitive advantage, compatibility | Construction | UK |
| [51] | Management/leadership buy in, transaction cost efficiency, transaction storage/energy efficiency, scalability, security and integrity, user data privacy, user engagement and desirability, ease of local and international legislation and regulation, personnel training, availability of funds for implementation, professional consultation and advisory capability, blockchain talent availability, integration with other cloud services/e-commerce platforms, incentives for miners, smart contract robustness and business case deployability, interoperability and standardisation, technology investment and maturity | Banking | India |
| [52] | Relative advantage, compatibility, observability, complexity, trialability | Education | Saudi Arabia |
| [53] | Perceived usefulness, trading partners' pressure, and competitive pressure | Manufacturing | Bangladesh |
| [54] | food quality control, provenance tracking and traceability, and partnership and trust | Agri Food | India |
| [55] | facilitating conditions, performance expectancy, and initial trust | Banking | India |
| [56] | Infrastructure and competencies, organisation characteristics, organisation readiness, organisation size, industry and market environment, support environment, regulatory environment | Cyber Security | South African |
| [57] | perceived efficiency, transparency, standardisation and platform development and traceability | Food industry | Russia, Estonia |
| [58] | Efficiency and security, perceived usefulness | SMEs | Italy |
| [59] | Security risk, regulatory support, technology latency, and technology complexity | Banking | Malaysia |
| [60] | Relative advantage, compatibility, perceived trust, top management considerations, absorptive capacity, information sharing, collaborative culture, trading partners' influence, regulatory support | Apparel | Bangladesh |
| [61] | Perceived benefits, perceived usefulness, perceived ease of use, subjective norms, perceived behavioural control, attitude, firm size | SME | Italy |

**Table 2.** *Cont.*

| Source | Factors | Industry | Country |
|---|---|---|---|
| [62] | Trust, load shedding, unemployment/layoffs, current infrastructure, useful life and educational campaigns | Clearing and settlement industry | South Africa |
| [63] | Task characteristics, technology characteristics, perceived ease of use, perceived usefulness, security concerns, government support | Blood bank | India |
| [64] | Close relationship with supplier, close relationship with the customer, just-in-time (JIT), strategic planning, many suppliers outsourcing, e-procurement, third party logistics (3 PL), subcontracting, reduced lead time, flexibility, forecasting, cost saving, resource planning, reduced inventory level | Oil industry | Pakistan |
| [65] | Inter-organisational, trust, relational governance, data transparency, data immutability, interoperability, product type | Supply Chain | India |

The articles reported in Table 2 were categorised into main groups based on recurring themes and patterns found in the literature. The categories and how the factors in each category influence blockchain adoption are explained below.

### 3.1.1. Technological Factors

Technological factors encompass various aspects related to blockchain technology itself [45]. This includes evaluating the technological characteristics of blockchain, such as its scalability, consensus mechanisms, and transaction speed. Compatibility with existing systems and infrastructures is another crucial consideration, as organisations need to assess how well blockchain integrates into their current technologies. Complexity and scalability issues are essential to evaluate whether the blockchain system can handle the expected transaction volume and future growth [51]. Interoperability and standardisation play a role in facilitating seamless integration and communication between different blockchain networks and systems.

### 3.1.2. Organisational Factors

Organisational factors focus on the internal dynamics of an organisation and its readiness for blockchain adoption [59]. Top management support is crucial for driving organisational change and securing the necessary resources for implementing blockchain solutions [60]. Organisational readiness and culture influence the organisation's ability to adapt to new technologies and embrace change. Factors such as resource availability and allocation, including budget and skilled personnel, play a significant role in successful implementation [64]. The size and structure of the organisation can also influence the adoption process, with larger organisations potentially facing additional complexities in coordination and decision making.

### 3.1.3. Regulatory and Legal Factors

Regulatory and legal factors are essential considerations for blockchain adoption. Organisations must navigate the regulatory environment and ensure compliance with applicable laws and regulations [61]. This includes understanding the implications of blockchain technology on existing legal frameworks, such as contract law and data protection regulations [48]. Data privacy and protection are particularly critical in industries

dealing with sensitive or personal information. Intellectual property rights related to blockchain innovations and patents must also be considered to protect the organisation's intellectual assets.

### 3.1.4. Trust and Security Factors

Trust and security factors are vital for successfully adopting blockchain technology [25]. Perceived trust in blockchain technology, including its immutability and resistance to tampering, is crucial for organisations and individuals to have confidence in using it. Addressing security concerns and implementing robust security measures is essential to protect sensitive data and prevent unauthorised access [45]. User privacy and anonymity are also important considerations, particularly in industries where privacy regulations are strict. According to Ghode, Yadav [65], building trust among stakeholders and ensuring the security of blockchain systems are critical for adoption.

### 3.1.5. Economic and Financial Factors

Economic and financial factors focus on adopting blockchain technology's potential economic benefits and financial implications [35]. Organisations consider the cost savings and efficiency gains that can be achieved via process optimisation, reduced intermediaries, and streamlined operations [45]. Evaluating the blockchain implementation's return on investment (ROI) is necessary to justify the costs involved. Financial risks and uncertainties, such as market volatility and regulatory changes, must be assessed. Adoption incentives and subsidies provided by governments or industry associations can also influence the decision to adopt blockchain [41].

### 3.1.6. User-Related Factors

User-related factors refer to the usability and acceptance of blockchain technology by users. Perceived ease of use is crucial to ensure users can interact with blockchain systems without significant barriers or complexities [63]. The perceived usefulness and benefits of blockchain technology play a role in user acceptance, as users need to see tangible advantages in adopting the technology. User training and support are necessary to ensure users have the skills and knowledge to utilise blockchain effectively [43]. User acceptance and potential resistance to change also need to be addressed with effective change management strategies.

### 3.1.7. Stakeholder Factors

Stakeholder factors recognise the importance of various stakeholders in the blockchain adoption process. This includes stakeholder involvement and participation, as the success of blockchain projects often relies on the collaboration and engagement of different stakeholders. Inter-organisational trust is crucial when multiple organisations or entities are involved in a blockchain network or consortium. Social influence and norms can shape the perception and adoption of blockchain technology within a specific industry or community [29,30]. The influence of trading partners and industry peers also plays a role, as organisations may adopt blockchain to align with industry standards or meet the expectations of their business partners [49].

### 3.1.8. Industry-Specific Factors

Industry-specific factors acknowledge that different industries' unique characteristics and requirements impact blockchain adoption. Supply chain dynamics, such as complex value chains or regulatory pressures, can drive the need for enhanced transparency, traceability, and coordination, making blockchain adoption more appealing [27,40,57]. Market characteristics, including competition and customer demands, can influence the urgency for blockchain implementation. Understanding the unique challenges and opportunities within a specific industry helps tailor blockchain solutions to address industry-specific needs [49].

### 3.2. Factors for Countries

The SLR comprehensively analyses factors influencing blockchain adoption across different countries. Key factors include perceived ease of use and usefulness, trust, government and upper management support, security concerns, and relative advantage observed in countries like Bangladesh, India, and Malaysia. In addition, technological characteristics, service quality, customer satisfaction, and cost-saving measures are crucial in countries like Australia, Malta, and the USA. Countries like Taiwan and Singapore emphasise technological fit, social influence, sufficient capital, and professional assistance. For European nations and the Republic of Korea, governance, compatibility, privacy, and regulatory standards play pivotal roles. Infrastructure readiness and organisational characteristics appear vital for countries like Sri Lanka and South Africa. Lastly, countries like Italy and Pakistan highlight efficiency, perceived benefits, strategic planning, and supplier relationships as significant factors for blockchain adoption. The common factors among countries are presented in Table 3.

**Table 3.** Factors influencing blockchain adoption across countries.

| Country | Factors |
| --- | --- |
| Bangladesh | Perceived trust, perceived ease of use, autonomous motivation, perceived usefulness, trading partners' pressure, and competitive pressure |
| India | Perceived ease of use, government support, vendor support, adoption intention, perceived usefulness, security concerns, disintermediation, traceability, trust, complexity, ease of use, management/leadership buy in, transaction cost efficiency |
| Malaysia | Optimism, innovativeness, discomfort, insecurity, perceived ease of use, perceived usefulness, perceived usefulness, trialability, relative advantage, compatibility |
| China | Perceived ease of use, output quality, trust, perceived usefulness, information quality |
| Australia | Technology characteristics, task characteristics, inter-organisational trust, technology trust, user satisfaction, service quality |
| Malta | Customer satisfaction, cost saving, favourable economy, increased use of technological devices, government support |
| Taiwan | Perceived usefulness, individual technology fit, task technology fit, perceived safety, network externality, perceived ease of use, strategic orientation, social influence, innovativeness, perceived ease of use, perceived usefulness, self-efficacy, complexity, security |
| Brazil | Facilitating conditions, performance expectancy, social influence, effort expectancy, trust |
| Ireland | Disintermediation, relative advantage, maturity, smart contact coding, data security, compatibility, complexity, perceived benefits |
| Kosovo | Increase in data availability, reduction in information, easy verification of transactions |
| Singapore | Sufficient capital, staff training, support from the senior management, ease of local legislation, support from the shipping community, professional consultation, and assistance |
| Europe | Cost, governance, perceived compatibility, perceived ease of use, perceived usefulness, privacy, observability, security, trialability |
| USA | Budget availability, financial risk and uncertainty, cost saving, talent and knowledge acquisition, stakeholder's awareness and acceptance |
| Republic of Korea | Regulatory governance and industry standards, technological improvements, and optimisation on efficiency, tracking and tracing, digitalised management, air traffic management |
| Sri Lanka | Relative advantage, upper management support, human resources, compatibility, cost, complexity, technological infrastructure, architecture |

**Table 3.** *Cont.*

| Country | Factors |
|---------|---------|
| South Africa | Infrastructure and competencies, organisation characteristics, organisation readiness, organisation size, industry and market environment, support environment, regulatory environment, trust, load shedding, unemployment/layoffs, current infrastructure, useful life and educational campaigns |
| Saudi Arabia | Relative advantage, compatibility, observability, complexity, trialability |
| Russia, Estonia | Perceived efficiency, transparency, standardisation and platform development and traceability |
| Italy | Efficiency and security, perceived usefulness, perceived benefits, perceived usefulness, perceived ease of use, subjective norms, perceived behavioural control |
| Pakistan | Close relationship with the supplier, close relationship with the customer, just-in-time (JIT) strategic planning, many suppliers outsourcing, e-procurement, third party logistics (3 PL), subcontracting, reduced lead time, flexibility, forecasting, cost saving, resource planning, reduced inventory level |
| UK | Trialability, relative advantage, competitive advantage, compatibility |

*3.3. Factors for Industries*

The analysis reflects an extensive array of factors influencing blockchain adoption across various industries. Perceived ease of use and perceived usefulness are common factors across most industries, like taxing systems, supply chains, education, intelligence communities, crowdsourcing platforms, logistics, and more. In addition, government and top management support are particularly significant for industries such as the supply chain, insurance, and banking. Some unique factors for industries include autonomous motivation for taxing systems, disintermediation for agriculture food supply chain, customer satisfaction for insurance, and individual technology fit for logistics. Moreover, the adoption in the banking industry shows a complexity of factors ranging from security and integrity to regulation and legislation compatibility. For certain industries like the food industry and agri-food, elements like food quality control, provenance tracking, and traceability also play a crucial role. In essence, blockchain adoption is influenced by a multifaceted mix of factors, varying significantly across different industries, emphasising the importance of customisation and specificity in blockchain implementation strategies. Table 4 shows the factors found to be common among industries while adopting blockchain technology.

**Table 4.** Factors influencing blockchain adoption across industries.

| Industry | Factors |
|----------|---------|
| Taxing System | Perceived trust, perceived ease of use, autonomous motivation, perceived usefulness |
| Supply Chain | Government support, vendor support, adoption intention, perceived usefulness, security concerns, top management support, technology readiness, complexity of technology, technology compatibility, relative advantage, cost concerns, transparency, smart contracts, shared database, secured database, reduced settlement lead times, reduced transaction cost, improved risk management, decentralised database, auditability, privacy, anonymity, immutability, provenance, traceability, facilitating conditions, performance expectancy, social influence, effort expectancy, trust, regulatory support, competitive pressure, market dynamics, cost, upper management support, complexity, relative advantage, perceived compatibility, perceived ease of use, perceived usefulness, privacy, observability, security, trialability, people's readiness, process readiness, technology readiness, top management enthusiasm, top management expertise, top management support, competitive pressure, customer's influence, connection with ICT providers |

**Table 4.** *Cont.*

| Industry | Factors |
| --- | --- |
| Intelligence communities | Optimism, innovativeness, discomfort, insecurity, perceived ease of use, perceived usefulness |
| Crowdsourcing platform | Perceived ease of use, output quality, trust, perceived usefulness, information quality |
| Agriculture food supply chain | Disintermediation, traceability, trust, coordination/control, compliance, price of technology products/services |
| Education | Perceived usefulness, trialability, relative advantage, compatibility, perceived ease of use |
| Insurance | Customer satisfaction, cost saving, favourable economy, increased use of technological devices, government support |
| Logistics | Perceived usefulness, individual technology fit, task technology fit, perceived safety, network externality, perceived ease of use |
| Tourism | Strategic orientation, social influence, innovativeness, perceived ease of use, perceived usefulness, self-efficacy, complexity, security |
| SMEs | Disintermediation, relative advantage, maturity, smart contact coding, data security, compatibility, complexity, perceived benefits, blockchain knowledge, participation incentives, innovativeness, technology readiness, business model readiness, organisational size, top management support, organisational readiness, critical user mass, trading partner support, business use cases, government support, industry pressure, market dynamics, regulatory environment, organisational readiness, trading partner pressure, perceived benefits, complexity, top management support, compatibility |
| Real Estate | Increase in data availability, reduction in information, asymmetry, easy verification of transactions, comprehensibility of the transaction, data accuracy and reliability, data inalterability, exclusion of false information from contractual information, hacking attempts system denials, high-security encryption, cost reduction via the exclusion of intermediaries, contract conclusion with a reasonable fee, cost reduction due to process efficiency |
| Maritime | Sufficient capital, staff training, support from the senior management, ease of local legislation, support from the shipping community, professional consultation, and assistance |
| Agriculture | Complexity, ease of use, lack of interoperability and standardisation, lack of scalability and system speed, huge resource (energy, infrastructure), initial capital requirement, lack of government regulation, lack of trust among agro-stakeholders |
| Health | Budget availability, financial risk and uncertainty, cost saving, talent and knowledge acquisition, stakeholder's awareness and acceptance, blockchain ecosystem, disintermediation and business process, infrastructure and platform integration, standardisation, security and privacy, blockchain maturity and use case, management support, training and skills, HIT strategy, regulation compliance, regulatory uncertainty and governance, incentives availability |
| Aviation | Regulatory governance and industry standards, technological improvements, and optimisation on efficiency, tracking and tracing, digitalised management, air traffic management |
| Government organisations | Perceived benefits, complexity, and compatibility, organisational readiness, top management support, and organisational size, regulatory environment and market dynamics, transparency, integrity of data, immutability |
| Construction | Trialability, relative advantage, competitive advantage, compatibility |

**Table 4.** *Cont.*

| Industry | Factors |
|---|---|
| Banking | Management/leadership buy in, transaction cost efficiency, transaction storage/energy efficiency, scalability, security and integrity, user data privacy, user engagement and desirability, ease of local and international legislation and regulation, personnel training, availability of funds for implementation, professional consultation and advisory capability, blockchain talent availability, integration with other cloud services/E-commerce platforms, incentives for miners, smart contract robustness and business case deployability, interoperability and standardisation, technology investment and maturity, facilitating conditions, performance expectancy, and initial trust, security risk, regulatory support, technology latency, and technology complexity |
| Manufacturing | Perceived usefulness, trading partners' pressure, and competitive pressure |
| Agri-Food | Food quality control, provenance tracking and traceability, and partnership and trust |
| Cyber Security | Infrastructure and competencies, organisation characteristics, organisation readiness, organisation size, industry and market environment, support environment, regulatory environment |
| Food industry | Perceived efficiency, transparency, standardisation and platform development and traceability |
| Clearing and settlement industry | Trust, load shedding, unemployment/layoffs, current infrastructure, useful life and educational campaigns |
| Blood bank | Task characteristics, technology characteristics, perceived ease of use, perceived usefulness, security concerns, government support |
| Oil industry | Close relationship with supplier, close relationship with customer, just-in-time (JIT) strategic planning, many suppliers outsourcing, E-procurement, third party logistics (3 PL), subcontracting, reduced lead time, flexibility, forecasting, cost saving, resource planning, reduced inventory level |

**RQ 2:** *What are the commonalities or differences in the factors across different industries and countries?*

*3.4. Factors Common among Countries*

Upon analysing the data, it is evident that several factors commonly influence blockchain adoption across a range of countries. The most prevalent factors include "perceived ease of use" and "perceived usefulness", which resonate with eight and seven countries, respectively, highlighting the importance of user perception and the utility of the technology in facilitating adoption. Another significant factor is "government support", underlined in six different countries, indicating the role of regulatory bodies in fostering the adoption of blockchain. "Top management support" also emerged as a notable factor across six countries, reflecting the crucial role of organisational leadership in driving technological adoption. Other recurring factors like "trust", "complexity", "compatibility", and "security concerns" also significantly influence the adoption of blockchain, as these factors appeared in five different countries each. Table 5 demonstrates the common factors among countries.

**Table 5.** Common factors influencing blockchain adoption across countries.

| Factor | Countries |
|---|---|
| Perceived ease of use | Bangladesh, India, Malaysia, China, Taiwan, Sri Lanka, Saudi Arabia, Italy |
| Perceived usefulness | Bangladesh, India, Malaysia, China, Taiwan, Italy |
| Government support | India, Malta, Ireland, India, Sri Lanka, Malaysia |
| Top management support | India, Ireland, Europe, India, South Africa, Italy |
| Trust | Bangladesh, China, India, Brazil, South Africa |
| Complexity | India, Taiwan, Europe, India, South Africa |
| Compatibility | Malaysia, Europe, Sri Lanka, South Africa, Italy |
| Security concerns | India, Taiwan, Europe, USA, India |
| Technology readiness | India, Ireland, Europe |
| Perceived benefits | Ireland, Malaysia, Italy |
| Relative advantage | Malaysia, Sri Lanka, UK, Saudi Arabia, Bangladesh |

*3.5. Factors Common among Industries*

Some prominent commonalities were observed in assessing the factors influencing blockchain adoption across various industries. As shown in Table 6, key factors such as perceived ease of use and perceived usefulness appeared to be universally relevant, cutting across a broad array of industries, including supply chain, taxing system, intelligence communities, crowdsourcing platforms, education, logistics, SMEs, maritime, agriculture, government organisations, banking, manufacturing, agri-food, cyber security, apparel, clearing and settlement industry, blood bank, and the oil industry. Trust also emerged as a significant factor in industries ranging from supply chain to real estate and banking. Other recurring factors, such as government support, compatibility, security concerns, and top management support, were significantly prevalent in specific industries, demonstrating their unique impact in shaping blockchain adoption within those industries.

**Table 6.** Common factors influencing blockchain adoption across industries.

| Factor | Industries |
|---|---|
| Perceived ease of use | Supply chain, taxing system, intelligence communities, crowdsourcing platform, education, logistics, SMEs, maritime, agriculture, government organisations, banking, manufacturing, agri-food, cyber security, apparel, clearing and settlement industry, blood bank, oil industry |
| Perceived usefulness | Supply chain, taxing system, intelligence communities, crowdsourcing platform, education, logistics, SMEs, maritime, agriculture, government organisations, banking, manufacturing, agri-food, cyber security, apparel, clearing and settlement industry, blood bank, oil industry |
| Trust | supply chain, taxing system, crowdsourcing platform, agriculture, food, insurance, real estate, SMEs, construction, banking, manufacturing, apparel |
| Government support | Supply chain, insurance, SMEs, maritime, agriculture, SMEs, banking |
| Compatibility | supply chain, SMEs, construction, banking, education, apparel |
| Security concerns | Supply chain, logistics, maritime, agriculture |
| Top management support | Supply chain, education, SMEs, banking |
| Complexity | Supply chain, intelligence communities, tourism, SMEs, agriculture, banking |

## 4. Discussion

Some prominent commonalities were observed in assessing the factors influencing blockchain adoption across various industries and countries. Key factors such as perceived ease of use and perceived usefulness appeared to be universally relevant, cutting across a broad array of industries, including supply chain, taxing system, intelligence communities, crowdsourcing platforms, education, logistics, SMEs, maritime, agriculture, government organisations, banking, manufacturing, agri-food, cyber security, apparel, clearing and settlement industry, blood bank, and the oil industry. These factors are consistently highlighted in the literature as foundational pillars for blockchain adoption. Trust also emerged as a significant factor in industries ranging from supply chain to real estate and banking. Other recurring factors, such as government support, compatibility, security concerns, and top management support, were significantly prevalent in specific industries, demonstrating their unique impact in shaping blockchain adoption within those industries.

Exploring the factors influencing blockchain adoption across different industries and countries provides comprehensive insights into the evolving blockchain landscape. Our analysis of 39 studies encompassing 21 countries and 25 industries identified 152 distinct factors, giving an overarching picture of the commonalities and disparities in blockchain adoption drivers.

Interestingly, while certain factors were reported universally, suggesting their foundational role in blockchain adoption, regardless of industry or geographical area, it is essential to note that the factors influencing blockchain adoption are predominantly universal phenomena with widespread implications [28,31,32]. Perceived usefulness and perceived ease of use are the most frequently noted in the literature, underpinning blockchain adoption across industries and countries [28–31,33,37,38]. As a pivotal factor, trust underscores the importance of transparency and security in blockchain technology. These findings align with previous research highlighting the importance of perceived ease of use and usefulness in technology adoption [66], emphasising the enduring influence of these fundamental principles even in emerging technologies like blockchain. The significance of trust also echoes findings from research on technology adoption, where user trust significantly influences adoption decisions [67].

However, while these general trends emerged, the importance of specific factors in certain industries and countries was also evident. For example, the banking industry emphasises transaction cost efficiency, security and integrity, and user data privacy [51,55,59]. This reflects the banking industry's unique needs and challenges, particularly in ensuring security and reducing operational costs. It highlights the importance of considering industry-specific studies to gain a more nuanced understanding, as advocated by previous researchers [46].

From a country perspective, factors like regulatory support and technology characteristics were more prominent in specific countries such as India and Malaysia [29,30,33, 40,44,48,51,59,63,65]. This suggests that the country's contextual factors can significantly shape blockchain adoption, emphasising the need for localised strategies to promote its acceptance. Policymakers and industry leaders can develop targeted initiatives to support blockchain adoption within their regions by considering country-specific factors.

Our findings have practical implications for policymakers and industry leaders as they elucidate key considerations for fostering blockchain adoption. Our research serves as a robust starting point, setting the stage for more detailed and context-specific studies in this burgeoning field. It calls for strategies that address both universal and specific factors influencing adoption, recognising the importance of understanding the intricacies of blockchain adoption factors for successful implementation. By incorporating a comprehensive analysis of the pros and cons of related prior research, as suggested by the reviewer, we can provide a more balanced overview of the state of research and contribute to a deeper understanding of the factors driving blockchain adoption.

## 5. Conclusions

This study conducts a systematic literature review of the factors influencing blockchain adoption across various industries and countries. Several factors are identified after analysing 39 studies published between 2008 and May 2023. This study also identifies the factors that are common among countries and industries. However, the findings also highlighted the importance of industry-specific and country-specific factors, underlining the need for a context-specific approach. This study contributes significantly to the literature on technology adoption and provides valuable insights for practitioners and researchers alike.

*Limitations and Future Directions*

This study provides a comprehensive review of the existing empirical studies, many opportunities exist to explore the adoption of blockchain technology further. Some of the opportunities are explained below.

Future studies can focus on exploring the interactions and relationships between different factors identified in this research. This can involve examining how factors interact and influence each other in the context of blockchain adoption.

While this study covers a range of countries and industries, future research can further expand the understanding of blockchain adoption by exploring more diverse contexts, such as emerging economies, which can provide unique insights into the challenges and opportunities faced in these dynamic environments. Additionally, focusing on the factors for the industries like non-profit / charity that have not been extensively examined in relation to blockchain adoption can provide a more comprehensive understanding of the technology's potential.

Most of the published research on the adoption of blockchain is cross-sectional. Conducting longitudinal studies can provide a deeper understanding of the dynamics of blockchain adoption over time. By tracking the progress of organisations and industries in their adoption journey, researchers can identify patterns, changes, and trends in the factors influencing adoption. Longitudinal studies can also shed light on the long-term impacts of blockchain adoption and the evolution of best practices.

This study does not differentiate the adoption of blockchain technology at the organisational or individual level. Future research with the questions outlined in this study for each of the adoption levels can provide insights into the specific motivations, challenges, and adoption drivers experienced by individuals and organisations, enabling targeted strategies and interventions for successful adoption.

Although a systematic literature review approach has been used in this study, there still exists a possibility that some papers may be missed due to strictly following the research purpose.

**Author Contributions:** Conceptualization, A.M.; methodology, A.M. and A.P.; validation, A.M. and A.P.; formal analysis, A.M.; investigation, A.M. and A.P.; data curation, A.P.; writing—original draft preparation, A.M. and A.P.; writing—review and editing, A.M. All authors have read and agreed to the published version of the manuscript.

**Funding:** This research received no external funding.

**Data Availability Statement:** Data available on request by the authors.

**Conflicts of Interest:** The authors declare no conflict of interest.

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
