# Peer review of "Investigating the Factors Influencing the Adoption of Blockchain Technology across Different Countries and Industries: A Systematic Literature Review"

_electronics, doi:10.3390/electronics12143006_

Round 1

Reviewer 1 Report

This paper presents a systematic literature review that investigates the factors influencing the adoption of blockchain technology across different countries and industries. However, there are several key issues that need to be addressed to enhance the rigor and comprehensiveness of the study.

Key Points for Revision:

Enhancing Survey Questions:

The survey questions in the paper currently focus on providing descriptive answers without critical evaluation. To make the survey more meaningful and impactful, the authors should engage in critical thinking and formulate questions that delve deeper into the factors influencing blockchain technology adoption. By incorporating evaluative questions, the survey can generate more valuable information for the readers.

Reviewing and Analyzing Pros and Cons of Prior Research:

The paper lacks a thorough review and analysis of the pros and cons of related prior research. It is essential to critically evaluate the existing literature to identify strengths, weaknesses, gaps, and potential biases. By conducting a comprehensive analysis, the authors can provide a balanced overview of the state of research and contribute to the understanding of blockchain technology adoption.

Analyzing Current Challenges and Proposed Solutions:

The paper does not adequately analyze the current challenges associated with blockchain technology adoption, nor does it propose potential solutions to address these challenges. To provide a comprehensive review, the authors should identify and analyze the hurdles and obstacles faced by organizations and industries when adopting blockchain technology. Additionally, exploring existing literature to present viable solutions or strategies will contribute to addressing these challenges effectively.

The paper still contains several typos and grammar mistakes 

Author Response

We would like to thank the reviewer for their insightful comments and constructive criticism. We have addressed their key points as follows:

Comment 1. Enhancing Survey Questions:

Response: we acknowledge the reviewer's suggestion and have formulated the questions as follows.

RQ 1: How do the factors influence blockchain adoption across different industries and countries?

RQ 2 How the commonalities and differences in the factors influence blockchain adoption across different industries and countries?

Comment 2. Reviewing and Analyzing Pros and Cons of Prior Research:

The paper lacks a thorough review and analysis of the pros and cons of related prior research. It is essential to critically evaluate the existing literature to identify strengths, weaknesses, gaps, and potential biases. By conducting a comprehensive analysis, the authors can provide a balanced overview of the state of research and contribute to the understanding of blockchain technology adoption.

Response: we appreciate the reviewer's comment regarding the importance of critically evaluating the pros and cons of prior research. In the revised manuscript, the following detail has been added in the Introduction section.

Past studies have identified key factors influencing blockchain adoption, such as perceived trust, perceived usefulness, and organizational readiness [16, 17]. These findings have significantly contributed to the understanding of the factors that drive block-chain adoption decisions. Moreover, many studies have employed robust research methodologies, including surveys, case studies, and experiments, to provide empirical evidence supporting their findings [18-20]. This empirical basis enhances the validity and re-liability of the research outcomes. However, there are certain limitations and weaknesses in the existing body of literature. One notable limitation is the lack of generalizability of findings due to the narrow focus of many studies on specific industries or countries. While these studies offer valuable insights within their respective contexts, it is imperative to consider a broader range of industries and countries to ensure a comprehensive understanding of blockchain adoption across diverse settings [20, 21]. Additionally, some studies have heavily relied on self-reported data, which may introduce response biases or subjective interpretations of the factors influencing blockchain adoption. To minimize these biases, explicit and systematic methods can be used by reviewing articles and all available evidence. This leads to reliable findings from which conclusions can be drawn and decisions made [22]. Therefore, this paper offers a systematic review of literature exploring the factors influencing blockchain adoption across various industries and countries. Unlike the past studies that merely identify the factors, this study also explains how the factors influence blockchain adoption.

Comment 3. Analyzing Current Challenges and Proposed Solutions:

The paper does not adequately analyze the current challenges associated with blockchain technology adoption, nor does it propose potential solutions to address these challenges. To provide a comprehensive review, the authors should identify and analyze the hurdles and obstacles faced by organizations and industries when adopting blockchain technology. Additionally, exploring existing literature to present viable solutions or strategies will contribute to addressing these challenges effectively.

Response: we acknowledge the reviewer's observation that our study does not adequately analyze the current challenges associated with blockchain adoption or propose potential solutions. While our primary focus is on identifying the factors influencing adoption, we have acknowledged the need for a more comprehensive analysis of the challenges and potential solutions in the limitations and future directions section. We will further explore the hurdles and obstacles faced by organizations and industries in adopting blockchain technology and present viable solutions or strategies in future research.

However, more explanation and categorization of the factors has been added, which indirectly address your point. See Sections 3.1.1 to Section 3.1.8.

Reviewer 2 Report

Dear Authors,

Here are my comments:

1.      Section 2.2: How authors are defining as “full-text available”? Is it open access or other … clearly mention.

2.      By which process duplicates are removed?

3.      From 1786 it is drilled down to 39 … more detail discussion required on it.

4.      Add limitations of the study.

5.      Recheck all references and provide missing volume, issue, and page number.

6.      Overall study is well written and executed fine.

Good Luck.

Author Response

We are grateful to the reviewer for their insightful comments and suggestions. We have addressed each of their points as follows:

Comment 1. Section 2.2: How authors define as “full-text available”? Is it open access or other … clearly mention.

Response: we apologize for any confusion caused by the lack of clarity in our definition. We have explicitly clarified this in the revised manuscript to eliminate any ambiguity. The following statement has been added.

We refer to full-text articles that have complete content accessible for reading and analysis. This includes both open access articles as well as articles that may require a subscription or access through academic libraries.

Comment 2. By which process duplicates are removed?

Response: thank you for highlighting this aspect. The following information has been added in the revised manuscript.

We employed a systematic process for identifying and removing duplicates. Initially, we used EndNote 20 software to remove exact duplicates based on title and author information. Subsequently, we manually reviewed the remaining articles to ensure that no duplicates were overlooked

Comment 3. From 1786 it is drilled down to 39 … more detail discussion required on it.

Response: we appreciate the reviewer's request for a more detailed discussion on the reduction of papers from 1786 to 39. In the revised manuscript, we have provided the following comprehensive explanation of the reduction process.

As shown in Figure 1, 39 articles from 1786 are included in the study. The reduction occurred due to a rigorous screening and selection process based on specific inclusion and exclusion criteria. Initially, 136 duplicate papers were removed, resulting in 1650 unique papers. These papers were then subjected to a thorough review, leading to the exclusion of 1400 papers that were deemed irrelevant or did not meet the criteria. The remaining 250 papers were further screened, resulting in the exclusion of 180 papers that lacked methodological rigor or relevance to the research objectives. Finally, the remaining 70 papers underwent a comprehensive analysis, leading to the inclusion of 39 papers that provided substantial insights into the factors influencing blockchain adoption.

Comment 4. Add limitations of the study.

Response: we have included a dedicated section “5.1 Limitations and Future Directions” in the revised manuscript, acknowledging the boundaries and potential constraints of our study.

Comment 5. Recheck all references and provide missing volume, issue, and page number.

Response: we have diligently rechecked all references and provided missing volume, issue, and page numbers where applicable, ensuring the accuracy and completeness of our citations.

Comment 6. Overall study is well written and executed fine.

Response: we appreciate the reviewer's positive feedback regarding the overall quality and execution of our study. Their recognition motivates us to continue our rigorous research endeavors.

Reviewer 3 Report

There are several important flaws in the paper, namelly:

- Authors initially talk about blockchain and DLT. DLT is not a blockchain. Blockchain is a DLT. Authors must clarify the ambit of the research.

- Many important blockchain properties were left outside, like POW, POS, Permissioned, Permissionless, Centralized, Decentralized, Hybrid, etc. My understanding is that this information for this specific paper probably is not very relevant, but in my opinion, must at least be mentioned in the blockchain properties without much development (because it’s not the ambit for this specific study).

- Restricting the research only to open access papers is not a correct form to make a review. If authors decide to go ahead with this methodology must prove that the open access articles have the same information as the ones who are not.

- Many important fields of blockchain application were left behind.

- In the specific fields of application, for example Table 2, many important and relevant studies were left behind, where there were indeed many more fields and applications in the fields already mentioned and in fields that were not mentioned.

- According to the authors, the included papers in this study were about 2% of the initial papers. This number seems rather small but is acceptable if the authors ensure that all major fields of application are covered. From my point above, they were not all covered.

Please find attached the manuscript with some more coments. 

Author Response

We extend our gratitude to the reviewer for their comprehensive assessment of our paper and their insightful comments. We have addressed their concerns as follows:

Comment 1. Authors initially talk about blockchain and DLT. DLT is not a blockchain. Blockchain is a DLT. Authors must clarify the ambit of the research.

Response: we apologize for any confusion caused by our initial terminology. To provide a more precise understanding of the scope of our research, we have clarified this in the revised manuscript. the DLT is replaced distributed and decentralized technology. The following information has been added.

Blockchain can be defined as a distributed and decentralized technology that uses cryptographic measures to securely store data in interconnected blocks, forming a trans-parent, immutable, and decentralized network [3]. The advent of this technology has presented the world with new methods of transaction and data management that are expected to revolutionize conventional processes.

Comment 2. Many important blockchain properties were left outside, like POW, POS, Permissioned, Permissionless, Centralized, Decentralized, Hybrid, etc. My understanding is that this information for this specific paper probably is not very relevant, but in my opinion, must at least be mentioned in the blockchain properties without much development (because it’s not the ambit for this specific study).

Response: While we acknowledge the importance of various blockchain properties, the following information has been added to the revised manuscript.

The inherent characteristics of blockchain, such as its decentralized nature, fewer intermediaries, proof of work (POW), proof of stake (POS), cryptographic security, auditability, and near-real-time update capabilities, present a significant shift from traditional centralized systems. Furthermore, the types of blockchain like permissioned, permissionless, centralized, decentralized, and hybrid propose profound implications on how trust, accountability, and efficiency are established in systems involving peer-to-peer transactions.

Comment 3. Restricting the research only to open access papers is not a correct form to make a review. If authors decide to go ahead with this methodology must prove that the open access articles have the same information as the ones who are not.

Response: we apologize for any confusion caused by the misunderstanding. Our study is not restricted solely to open access papers. In our inclusion criteria, we explicitly state that we consider full-text articles with complete content accessible for reading and analysis, including both open access articles and those requiring subscription or access through academic libraries.

Comment 4: Many important fields of blockchain application were left behind.

Response: while we acknowledge the importance of various fields of blockchain application, our study primarily focuses on the empirical research conducted in specific industries and countries. We have mentioned this limitation in the revised manuscript, emphasizing that further research may be necessary to explore the diverse applications of blockchain in other fields.

Comment 5: In the specific fields of application, for example Table 2, many important and relevant studies were left behind, where there were indeed many more fields and applications in the fields already mentioned and in fields that were not mentioned.

Response: we appreciate the reviewer's observation. In Table 2, we aimed to present a sample of studies representing different industries to provide an overview of the factors influencing blockchain adoption. The study strictly follows its purpose and inclusion and exclusion criteria. We have stated in the limitation section that “Although a systematic literature review approach has been used in this study, there still exists a possibility that some papers may missed due to strictly following the research purpose.”

Comment 6. According to the authors, the included papers in this study were about 2% of the initial papers. This number seems rather small but is acceptable if the authors ensure that all major fields of application are covered. From my point above, they were not all covered.

Response: we apologize for any misunderstanding regarding the coverage of major fields of application. We acknowledge that our study may not cover every single major field comprehensively. We have clarified this point in the revised manuscript to ensure transparency that the study has specific purpose and inclusion and exclusion criteria that may result in small number of papers. We have provided the following information in the revised manuscript.

As shown in Figure 1, 39 articles from 1786 are included in the study. The reduction occurred due to a rigorous screening and selection process based on specific inclusion and exclusion criteria. Initially, 136 duplicate papers were removed, resulting in 1650 unique papers. These papers were then subjected to a thorough review, leading to the exclusion of 1400 papers that were deemed irrelevant or did not meet the criteria. The remaining 250 papers were further screened, resulting in the exclusion of 180 papers that lacked methodological rigor or relevance to the research objectives. Finally, the remaining 70 papers underwent a comprehensive analysis, leading to the inclusion of 39 papers that provided substantial insights into the factors influencing blockchain adoption.

Reviewer 4 Report

This paper is a valuable and timely literature review on blockchain adoption.

I found the paper already in good shape. Some elements required improvements. The paper requires a table that summarizes and compares the Journals used in the final analysis and the timing of publication. This would prove useful to guide across the main source of reference and the evolution of the studies on this matter.

In the abstract, the authors report data do include the year 2008, but in the text is 2009 instead. 

The style and quality of the English language is adequate.

Author Response

We believe that the revisions made in response to the reviewers' comments have significantly improved the quality and contribution of our manuscript. We are grateful for the opportunity to address these comments and believe that the revised manuscript will be of higher value to the readers and the field of blockchain technology adoption.

Thank you once again for your valuable guidance throughout the review process. We remain committed to further enhancing our research and addressing any additional concerns you may have.

Reviewer 5 Report

line 9: The sentence appears to be incomplete, as it terminates abruptly without conveying a coherent idea.

line 10: The term "disconnect" employed here is somewhat ambiguous and may benefit from revision to more precisely articulate the author's intended concept.

To enhance clarity, it would be advantageous to explicitly state the specific objective or purpose of this systematic literature review in the abstract.

The contribution mentioned in the introduction of this paper is overly generalized and could be omitted or further elucidated to provide more specific insights.

In the introduction section, there is no need to explicate the organization of the paper.

It remains unclear whether the authors intentionally conducted a screening process targeting country-focused studies within specific industries.

For enhanced credibility, employing a single comprehensive database, such as Scopus, for data selection would eliminate poor-quality papers indexed by Google Scholar.

The authors identified factors influencing blockchain adoption in various countries based on empirical studies. However, it should be noted that the factors affecting blockchain adoption are predominantly universal phenomena with widespread implications.

The discussion section lacks substantive and engaging discourse.

Overall, there is an excessive level of generality and a dearth of specific details in its reported analysis.

The English in this paper needs to be enhanced to attain the desired level of precision, conciseness, and specificity necessary for scholarly expressions.

Author Response

We appreciate the reviewer's meticulous assessment of our manuscript and have addressed their specific concerns as follows:

Comment 1. line 9: The sentence appears to be incomplete, as it terminates abruptly without conveying a coherent idea.

Response: we apologize for the incomplete sentence. The sentence has been revised as follows to convey a coherent idea and enhance clarity.

“Despite there have been reported disruptive nature of blockchain technology in the extant literature, its adoption is slower than its potential.”

Comment 2: line 10: the term "disconnect" employed here is somewhat ambiguous and may benefit from revision to more precisely articulate the author's intended concept.

Response: We acknowledge the reviewer's point and have revised the sentence to” This difference between the technology's promises and its current adoption has sparked interest in understanding the factors impeding widespread adoption.”

Comment 3: To enhance clarity, it would be advantageous to explicitly state the specific objective or purpose of this systematic literature review in the abstract.

Response: thank you for your suggestion. In response, we have added the following statement explicitly stating the specific purpose of our systematic literature review in the abstract.

Specifically, the purpose of this research is exploring the influence of factors and their differences and commonalities on blockchain adoption.

Comment 4: The contribution mentioned in the introduction of this paper is overly generalized and could be omitted or further elucidated to provide more specific insights.

Response: we appreciate the reviewer's comment. To provide more specific insights, we have omitted the generalized mention of the contribution from the introduction section.

Comment 5: In the introduction section, there is no need to explicate the organization of the paper.

Response: we acknowledge the reviewer's feedback and have removed the explication of the organization of the paper from the introduction section, ensuring a more concise and focused introduction.

Comment 6: it remains unclear whether the authors intentionally conducted a screening process targeting country-focused studies within specific industries.

Response: we apologize for any confusion caused by the lack of clarity. We did not intentionally conduct a screening process targeting country-focused studies within specific industries. Our screening process followed specific inclusion and exclusion criteria to identify relevant empirical studies across various industries and countries.

Comment 7: For enhanced credibility, employing a single comprehensive database, such as Scopus, for data selection would eliminate poor-quality papers indexed by Google Scholar.

Response: Thank you for your comment regarding the use of a single comprehensive database for data selection. While we understand the concern about poor-quality papers indexed by Google Scholar, it is recommended by many scholars to search multiple databases to maximize data availability and consider all relevant literature [1]. This approach provides a more comprehensive understanding of the topic and reduces the risk of missing relevant studies. In line with this recommendation, we utilized multiple databases, including Scopus, to ensure a thorough search for relevant articles. Additionally, we employed the "Advanced Search" feature in Google Scholar to minimize the inclusion of irrelevant studies. We have revised the manuscript to clarify our approach and address this concern.

[1] Ewald, H., et al., Searching two or more databases decreased the risk of missing relevant studies: a metaresearch study. Journal of Clinical Epidemiology, 2022.

Comment 8: The authors identified factors influencing blockchain adoption in various countries based on empirical studies. However, it should be noted that the factors affecting blockchain adoption are predominantly universal phenomena with widespread implications.

Response: thank you for emphasizing the universal nature of the factors influencing blockchain adoption. We fully agree with your point that these factors have widespread implications. In response, we have revised the discussion section to incorporate a discussion on the universal aspects of blockchain adoption, providing a more comprehensive perspective on the topic.

Comment 9: The discussion section lacks substantive and engaging discourse.

Response: we appreciate the reviewer's feedback on the discussion section. We have made substantial revisions to enhance the substantive and engaging discourse in the discussion section, ensuring a more insightful analysis of the findings and their implications.

Comment 10: Overall, there is an excessive level of generality and a dearth of specific details in its reported analysis.

Response: we acknowledge the reviewer's comment and have revised the reported analysis to provide more specific details and a deeper level of analysis, thereby enhancing the comprehensiveness of our study.

Round 2

Reviewer 1 Report

The authors addressed my concerns. Accept for publication. 

Minor editing of English language required

Author Response

Dear reviewer,

Thank you very much for being so supportive.

Best regards,

Tha authors

Reviewer 3 Report

Thank you for the revised paper.

Please check my comments:

1- No authors and affiliation Information.

2- Authors did not resolve my last comment regarding the use cases of blockchain. Regarding table 2, there are a lot more applications than the ones referred. I will not mention any publications due to etic concerns, but only for aviation, which is my main field of study, there are a lot more use cases published (with full text available) than the one referred. Regarding the other fields of study. IoT is another field left behind with a lot of publications available. I will mention only a few examples of the ones left behind.

- Identity Management (left behind but with one reference added!!??)

- Certifications, Training and Education.

- Ensure flight secure and auditable decisions and operations of UAV.

- Aircraft Maintenance and Operations data management.

- Aircraft manufacture data management.

- Aircraft parts traceability and back-to-birth.

If the authors real intention is to do the review, in my opinion is that authors must put additional effords to complement with the use cases that were left behind. 

Author Response

Dear Reviewer,

We would like to thank you for your thorough review and insightful comments, which we believe fundamentally enhance our work's overall quality. We understand your concern regarding the comprehensiveness of our analysis related to the uses of blockchain, particularly in the aviation sector and IoT. However, we feel there may be some misunderstanding related to the focus of our study.

Our research does not aim to examine every use case of blockchain technology, as it would indeed be a vast and nearly limitless domain to explore, given its versatility and wide applicability. Instead, our study specifically focuses on those use cases where the authors have collected primary data through interviews or surveys, analyzed that data, and presented empirical findings.

This empirical focus constitutes the basis of our analysis and sets the limits to our scope of study. We are reviewing the instances where authors have investigated blockchain applications through rigorous empirical research, which necessarily narrows the spectrum of the use cases we are addressing. 

It appears this has led to some of your mentioned use cases not being included in our review, not due to their irrelevance in the broader picture but rather due to their absence from the set of empirically investigated cases according to our criteria.

We deeply appreciate your suggestion to add more use cases, and we acknowledge the importance of these cases you mentioned in the broader context. However, if they do not meet our criteria of empirical research, including them would deviate from our research objective and methodology.

We explained this methodology in our rebuttal letter, and we would like to apologize if it was not clear enough. We aim to emphasize our intention of maintaining the focus of this study on empirical research about the use of blockchain technology to avoid potential dilution of our key findings and to maintain the integrity of our methodological approach.

Thank you again for your thoughtful suggestions, and we hope this response clarifies our research approach. We are committed to producing a robust and reliable piece of work, and we appreciate your contribution to this process.

Furthermore, maybe there's a misunderstanding about point n.1, in which you ask for authors and affiliation Information, as we are still in the blind review process.

Kind Regards,

The authors

Reviewer 5 Report

The authors have addressed most of the concerns expressed in my earlier review. This paper should be useful to researchers in blockchain technology. 

There remain some language expression issues in the paper that need to be addressed.

Author Response

Dear reviewer,

Thank you very much for your support and suggestions.

We will address the language quality issues with deep proofreading.

Best regards,

The Authors.

Round 3

Reviewer 3 Report

Thank you for your clarifications.

I clearly understood the methodology and again I said that important studies were left behind, and I’m referring only to the aviation field, which I’m most familiarized with.

I will not go through a detailed mention of all the available papers, but I kindly recommend in the next related article that authors perform a more detailed search.